# A Review of Recent Chitosan Anion Exchange Membranes for Polymer Electrolyte Membrane Fuel Cells

**DOI:** 10.3390/membranes12121265

**Published:** 2022-12-14

**Authors:** Vijayalekshmi Vijayakumar, Sang Yong Nam

**Affiliations:** 1Research Institute for Green Energy Convergence Technology, Gyeongsang National University, Jinju 52828, Republic of Korea; 2Department of Materials Engineering and Convergence Technology, Gyeongsang National University, Jinju 52828, Republic of Korea

**Keywords:** anion exchange membrane, biopolymer, chitosan, composites, fuel cell

## Abstract

Considering the critical energy challenges and the generation of zero-emission anion exchange membrane (AEM) sources, chitosan-based anion exchange membranes have garnered considerable interest in fuel cell applications owing to their various advantages, including their eco-friendly nature, flexibility for structural modification, and improved mechanical, thermal, and chemical stability. The present mini-review highlights the advancements of chitosan-based biodegradable anion exchange membranes for fuel cell applications published between 2015 and 2022. Key points from the rigorous literature evaluation are: grafting with various counterions in addition to crosslinking contributed good conductivity and chemical as well as mechanical stability to the membranes; use of the interpenetrating network as well as layered structures, blending, and modified nanomaterials facilitated a significant reduction in membrane swelling and long-term alkaline stability. The study gives insightful guidance to the industry about replacing Nafion with a low-cost, environmentally friendly membrane source. It is suggested that more attention be given to exploring chitosan-based anion exchange membranes in consideration of effective strategies that focus on durability, as well as optimization of the operational conditions of fuel cells for large-scale applications.

## 1. Introduction

Eco-friendly energy sources as alternatives to fossil fuels have attracted significant interest owing to their ability to solve the problems associated with environmental pollution and power shortages [1,2]. Polymer electrolyte membrane fuel cells (PEMFCs) as an energy conversion device stand out in various fields, including stationary as well as portable electronics, transportation, etc. [3,4,5]. Based on the properties of the polymer membrane, there are two types of PEMFCs: proton exchange membrane fuel cells (PEMFCs) in which a polymer electrolyte membrane (PEM) is responsible for proton migration from an anode to a cathode, and anion exchange membrane fuel cells (AEMFCs), which use an anion exchange membrane (AEM) to conduct hydroxyl ions. The attractive features possessed by AEMFCs, such as faster fuel cell reaction kinetics, efficient water management, lower methanol permeability, and the usage of non-noble metal catalysts, indeed make them more attractive than proton exchange membrane fuel cells. However, the lower ionic conductivity and poorer electrochemical, thermal, alkaline, and mechanical stability associated with the anion exchange membranes resulted in far inferior cell performance for AEMFCs than PEMFCs [6,7]. Therefore, there is a significant interest in developing an ideal anion exchange membrane with long-term mechanical, thermal, as well as chemical stability and good electrochemical properties. Typical AEMs are made of aromatic polymers grafted with cationic groups, but the grafting process generally undergoes chloromethylation following the Menshutkin reaction involving the usage of carcinogenic chloromethyl methyl ether, which is not suitable for large-scale preparation of AEMs [8]. Therefore, developing membranes using green processes is a hot topic in the field of AEMFCs.

The use of the most abundant low-cost non-toxic biopolymers found in nature, especially chitosan and its derivatives, as an alternative material for AEMs has gained widespread attention because of its excellent film-forming properties and high chemical stability [9,10,11,12]. However, chitosan, an abundant natural cationic polysaccharide, had some limitations in its pure form, such as poor mechanical stability and poor conducting properties, which led to chemical modifications for better performance. The presence of -NH_2_ and -OH groups in its structure facilitates easy modification and avoids the carcinogenic chloromethylation step [13,14,15,16]. The review discusses the performance of chitosan-derived membranes for anion exchange membrane fuel cells in the recent past (2015–2022). Various strategies, such as the preparation of structurally modified hydrophilic/hydrophobic copolymers, blending chitosan with other polymers, chemically modifying chitosan, and creating membranes based on organic/inorganic hybrids, are included.

## 2. Chitosan Membranes

### 2.1. Structural Modification

Anion exchange membrane’s basic structure consists of a hydrophobic polymer backbone and positively charged hydrophilic functional groups such as quaternary ammonium, imidazolium, benzimidazolium, phosphonium, guanidinium, metal cations, etc. Various kinds of AEMs were reported, but there are still significant challenges to commercialization, such as poor hydroxyl ion conductivity and alkaline instability of the cationic groups under high pH conditions and elevated temperatures [17]. Therefore, it is necessary to develop an ideal AEM that possesses long-term thermal and alkaline stability as well as high ionic conductivity and mechanical properties. Quaternary ammonium cations, the commonly used cationic species, are susceptible to degradation via direct nucleophilic substitution (S_N_2), Hofmann elimination reaction (E2), or Ylide formation (Y), as shown in Figure 1 [18]. To explore novel AEMs with high conductivity and chemical stability at elevated temperatures for long-term fuel cell performances, the imidazolium-based one has evoked great interest due to its enhanced alkaline stability. It is worth pointing out that the resonance effect of the conjugated imidazole ring could diminish the interaction of imidazolium cations with hydroxyl ions and improve alkaline stability. Ryu et al. incorporated 4-pyridine carboxaldehyde derivatives into chitosan, which was then copolymerized with vinyl imidazole to improve its mechanical, chemical, and conductive properties. The low volatility, good chemical and thermal stabilities, as well as the vast electrochemical dynamic range of these materials, improved the membrane’s conductivity. An alkaline direct methanol fuel cell using the fabricated membrane produced an output peak power density of 10.42 mW cm^−2^ at a corresponding current density of 28.76 mA cm^−2^ [1]. 

As the diffusion coefficient of H^+^ is four times greater than that of OH^−^, a four-fold increase in ion exchangeable groups is required, but high ion exchange capacity (IEC) can lead to excessive swelling and loss of mechanical properties. Many scholars adapted glutaraldehyde, ethylene glycol diglycidyl ether, 1,4 dichlorobutane, epichlorohydrin, etc. to solve this problem through chemical cross-linking [19,20]. Numerous studies on cation type and cross-linker have been presented, including a study by Wan et al. in which quaternized chitosan derivatives synthesized using glycidyl trimethylammonium chloride and crosslinked with ethylene glycol diglycidyl ether were investigated. Anchoring two epoxy groups of the crosslinker onto two amino groups in different chitosan chains through covalent linkage forms the crosslinked network structure. They concluded that increasing cation charge density, as well as crosslink density modulated membrane performance [11]. After crosslinking with glutaraldehyde, a membrane based on chitosan modified with a reactive cationic dye synthesized using 1-aminoanthraquinone, dimethyl propylene diamine, cyanuric chloride, and diethyl sulfate as reactants achieved hydroxyl ion conductivity of 4.59 × 10^−3^ S cm^−1^ at 30 °C and alkaline stability of 300 h at 80 °C in 8 M KOH solution. In this work, the chemically reactive dye was grafted onto the chitosan matrix through a substitution reaction between the dye’s chlorotriazine groups and the chitosan’s -OH/-NH_2_ groups at the appropriate temperature to provide hydroxide-conductive charge carriers. Focusing on the major concern associated with excess swelling in the presence of hydrophilic components in both materials, chemical crosslinking with glutaraldehyde was also adopted to balance the conductive as well as mechanical performances of the AEMs. The observed activation energy of ion migration (8.27 kJ mol^−1^) suggested the co-existence of both Grotthus and vehicle mechanisms, with the former dominanting [21]. AEM (CS-PY-DL_x_) prepared by using Schiff base functionalized chitosan synthesised using by 4-pyridine carboxaldehyde, 1,4 dichlorobutane as a crosslinker, and iodomethane as quaternization reagent also achieved an ionic conductivity of 7.08 × 10^−3^ S cm^−1^ at 70 °C and retained the value of about 79.7% after 120 h exposure in 3 M KOH solution at 80 °C and revealed its excellent alkali stability. The delocalized π-bond conjugated system in the pyridyl group protects the quaternary ammonium, and the crosslinked network improves the methanol resistance and mechanical as well as alkaline stability of the system. The three-dimensional network structure becomes more dense, and the regularity of molecular chains increases with an increase in crosslink density, which results in a hindrance to ionic hopping transport and a reduction in conductivity at high crosslinking degrees [22]. The synthesis scheme and molecular structure of the membrane are illustrated in Figure 2.

Taking advantage of the good alkaline stability of imidazole as well as improving conductivity by providing ionic clusters Wang et al. synthesized a bi-imidazolium functionalized chitosan-based AEM using butanediyl-1,4-bis (N-dodecyl imidazole bromide) ionic liquid monomer (Figure 3), which showed an anionic conductivity of 41.9 mS cm^−1^ at 80 °C. The long hydrophobic side chain (dodecyl) on imidazole promoted the formation of ionic clusters (N^+^), which can facilitate the formation of broad interconnected ionic channels for conduction, and N^+^ active sites dispersed in imidazole rings weaken the interaction with hydroxide ions, thereby extending the alkaline stability of AEMs. An icrease in ionic liquid concentration boosts the number of N^+^ cationic sites that facilitate more water absorption. The water in the membranes can be classified into free and bound water; the former can affect OH^−^ ion transport. The membrane with an ionic liquid fraction of 15 wt% displayed the highest hydroxide conductivity and stability in high pH solutions especially in a 1 M KOH methanol solution. Glutaraldehyde was used as the crosslinking agent [17]. In 2015, Song et al. published their work on AEMs prepared by thermal and chemical crosslinking of chitosan modified with 1-ethenyl-3-methyl-1H-imidazolium chloride polymer and 1-ethenyl-2-pyrrolidone (CS/EMImC-Co-EP). The membranes showed a hydroxide ion conductivity of about 0.01 S cm^−1^ at 80 °C and high stability without losing their integrity and conductivity during 300 h of exposure in 8 M KOH at 85 °C. The five membered heterocyclic ring structure offered a π-conjugated structure and steric hindrance, thereby reducing the possibility of S_N_2 substitution and Hofmann elimination reactions. Membrane electrode assembly fabricated with CS/EMImC-Co-EP-OH^−^ membrane with a blend ratio of 1:0.5 (by mass) showed a maximum power density of 21.7 mW cm^−2^ and an open circuit voltage of 0.92 V at room temperature and ambient pressure [23].

The unique hole structure and electronegativity of crown ether have attracted researchers’ attention to its potential use in the field of fuel cells to increase the ion migration rate. The electron negativity of crown ether can complex metal ions to increase ionic conductivity as well as chemical stability. Owing to its size, K^+^ is the most suitable and stable in 18 crown 6. Zheng et al. reported a kind of AEMs with chitosan-crown ether membranes prepared by the Schiff reaction between chitosan and dibenzo-18-crown-6 (DB_18_C_6_) for alkaline fuel cells. The highest hydroxide ion conductivity of 52.18 mS cm^−1^ achieved at 70 °C and alkaline stability after exposure in 6 M KOH solution for 480 h reveal that the maximum conductivity degradation is only 5%, indicating excellent chemical stability [7].

An interpenetrating network has been used to prepare AEMs with improved mechanical as well as chemical stability. A high-strength AEM with a full interpenetrating network (IPN) based on quaternized chitosan, polyacrylamide, and polystyrene achieved a low methanol permeability, less swelling, and good tensile stress. The conductivity of about 90% was maintained even after being soaked in 1 M KOH for 120 h and 10 M KOH for 50 h at room temperature. Conductivity was improved from 0.006 S cm^−1^ to 0.013 S cm^−1^ as QCS increased from 60% to 90% in IPN but needs to improve to be used in AEMFC [24]. Semi-interpenetrating chitosan (CS)-based AEM prepared by incorporating poly-(acrylamide-co-diallyl dimethylammonium chloride) (PAADDA) and linear structured poly-bis(2-chloroethyl)ether-1,3-bis [3-(dimethylamino)propyl] urea copolymer (PUB) as charge carriers through solution casting technique supported by thermal and chemical crosslinking methods reached a conductivity up to 16 mS cm^−1^ at ambient temperature and 31.2 mS cm^−1^ at 80 °C. The existence of cyclic structure along with quaternary ammonium group in PAADDA and space sharing of an electron pair from non-quaternized nitrogen on PUB effectively inhibited the degradation of quaternary ammonium groups. Simultaneously, the CS/PAADDA/PUB membrane demonstrated good alkaline resistance as well as oxidative stability when treated in 8 M KOH solution at 60 °C for 320 h and 30% H_2_O_2_ solution at room temperature for 120 h. A maximum power density of 38.1 mW cm^−2^ at a peak current density of 73.2 mA cm^−2^ in an H_2_/O_2_ fuel cell at room temperature was achieved by the fabricated membrane electrode assembly (MEA) (Figure 4). In the author’s point of view, improvements in many aspects, such as composition and crosslinking manners as well as MEA fabrication techniques, have to be enhanced to strengthen the cell’s performance [25].

### 2.2. Composite Membranes

It is well known that the hydroxyl ions are transported and diffused in an alkaline medium through two mechanisms: the Grotthuss and vehicular mechanisms in those waters play a significant role. Water clusters benefit the OH^−^ ion conduction in the membrane, and therefore, an appropriate amount of water is required to improve the conductivity. However, an excess of water dilutes the charge carrier, causing a decline in conductivity as well as a loss of dimensional stability, which has an adverse effect on the membrane’s durability [26]. An increase in quaternary ammonium groups in membrane structure owing to high ion exchange capacity (IEC) as well as hydration degree results in unwanted swelling and thereby a reduction in membrane stability. To solve the trade-off between ionic conductivity and mechanical stability, various approaches have been reported. Organic-inorganic hybrid composites, owing to the combination of properties of both organic and inorganic components, can offer ion-conducting sites and control membrane swelling, thus boosting the mechanical properties. Different inorganic materials, including nanoparticles, one-dimensional nanotubes, and two-dimensional nanosheets, have been utilized to improve the performance of composite AEMs. Cationic nanofibers, owing to their one-dimensional, long-range, continuous hydroxide ion conducting pathways, are often used as the ion conducting media of AEMs. A quaternized silica-coated poly(vinylidene fluoride) (QSiO_2_@PVDF) electro-spun nanofiber mat fabricated using general polydopamine-assisted sol-gel method filled with quaternized chitosan (QCS) exhibited high ion conductivity of about 0.041 S cm^−1^ at 80 °C, which is 1.3 times greater than that of pure QCS membrane. When used in a single alkaline fuel cell, the design of a mechanically supported fibre core together with a high concentration ion transport surface offers high-performance AEMs with a power density of 98.7 mW cm^−1^. The durability test resulted in only a 4.2% reduction in performance after a 100 h chronoamperometry test, proposing that anion nanofiber substrates with an inner hydrophobic fibre core and an outer hydrophilic surface design are good for the preparation of highly stable and conductive AEMs [27]. 

Blending synthetic polymers like polyvinylchloride (PVC) with natural polymers like chitosan after quaternization, while avoiding carcinogenic chemicals for both chloromethylation and bromination, is a facile route to prepare an anion-conducting membrane. Hari Gopi et al. developed a series of membranes using dual quaternization agents such as hexadecyltrimethylammonium bromide (HDT) and 2,3,5-triphenyl tetrazolium chloride (TPTZ), followed by crosslinking using chemical and thermal methods. According to the conductivity and stability studies, it was found that the polymer quaternized using HDT is a viable alternative for use as an electrolyte in an anion exchange polymer electrolyte membrane fuel cell, whereas the peak power density achieved (15 mW cm^−2^) indicates that further work is required to optimize other parameters such as effect of cell temperature, ionomer concentration in catalyst ink, etc. [28]. Glycidyltrimethylammonium chloride (GTMAC), quaternized chitosan (QCS), quaternized poly vinyl benzyl chloride (QVBC), and polysulfone (PSF) blended with 1,4-dibromobutane showed an ionic conductivity of 49.6 mS cm^−1^ and 130 mS cm^−1^ at 25 °C and 70 °C, respectively. The hydrophilic/hydrophobic phase separation owing to the presence of two different phases resulted in well-connected ion channels that accelerate hydroxide ion transport in the membrane [29]. A blend of N, N-dimethyl chitosan (DMC) and 1,4-diazoniabicycle [2.2.2]octane functionalized polysulfone after crosslinking with 1,4-dibromobutane also exhibited good dimensional stability as well as an ion conductivity of 54 mS cm^−1^ and 94 mS cm^−1^ at 25 °C and 70 °C, respectively [30].

Organic-inorganic hybridization is a promising method due to the synergistic effect of both the organic and inorganic components. Analogous to graphene, molybdenum disulfide (MoS_2_) also shows extensive attraction in fuel cell membrane preparation due to its 2D lamellar structure and benefits to ion transport. A homogeneous dispersion of MoS_2_ in a quaternized polyvinyl alcohol (QPVA)/chitosan blend could enhance the membranes’ mechanical properties and reduce methanol permeability owing to the strong interaction among the MoS_2_ layer and polymer chains [31]. Zhou et al. used chitosan, an ionised organic compound named N-Benzyl-N,N-dimethyl-3-{[2-methyl-1,3-dioxo-2,3-dihydro-1H-benzo(de)isoquinolin-6-yl]amino}propan-1-aminium hydroxide ((QAIM)OH), as well as hydroxylated multiwalled carbon nanotubes (MWCNTs-OH) to cast membranes supported by the glutaraldehyde (GA) crosslinking process. The composite membranes showed a peak ionic conductivity of 5.66 mS cm^−1^ at room temperature and a power density of 31.6 mW cm^−2^ in the H_2_/O_2_ fuel cell system at room temperature [26]. AEM prepared using chitosan (CS), magnesium hydroxide (Mg(OH)_2_), and graphene oxide (GO) with benzyl trimethylammonium chloride (BTMAC) as the hydroxide conductor (Figure 5) also showed a high conductivity of 142.5 mS cm^−1^ at 40 °C and a peak power density of 68.6 mW cm^−2^ at 80 °C. After enzymatic crosslinking with dodecyl 3,4,5-trihydroxybenzoate having a C-10 alkyl chain (DTHB), improvements in mechanical properties as well as hydrolytic stability were noticed, but conductivity and power density were 92.8 mS cm^−1^ at 40 °C and 51.3 at 80 °C, respectively. The presence of oxygenic functional groups such as hydroxyl, epoxy, and carboxylic groups located at the edge of graphene oxide governs anion transfer, and hydrophobic regions in the aromatic ring (sp^2^ carbon layer) help to improve fuel crossover resistance across the membrane [32].

Layered double hydroxides (LDHs), a class of lamellar anionic clays comprised of positively charged metal hydroxide layers with negatively charged ions and water molecules in the interlayer, have been investigated as a promising anion conductor for AEMFC applications. This positively charged feature, along with hydroxyl groups in the interlayer and water molecules, facilitated the formation of abundant hydrogen bonding along the 2D surface and promoted OH^−^ ion conduction through diffusion or the vehicle mechanism [8]. However, the conventional methods generally resulted in bulk LDH with serious layered stacks because of the ab-face stacking and strong van der Waals interactions between nanosheets during the drying process. LDH with single-layer nanosheets or a smaller number of layers and their good distribution are critical to achieving high overall performance of the membranes. Therefore, researchers adopted surface modification techniques with cation groups to promote the dispersion and ion exchange capacity without sacrificing the dimensional stability of hybrid AEMs. Hu et al. reported glycine betaine intercalated LDH (B-LDH) as a new multi-functional additive to modify the QCS/PVA blend, and the presence of an organic anion interlayer promoted dispersion and interfacial adhesion in the matrix, thereby improving the mechanical strength (maximum TS: 23.6 MPa, elongation: 51.4%). B-LDH can also function as a physical crosslinking point to hinder the fracture of the composite. The addition of 5% B-LDH showed an anion conductivity of 35.7 mS cm^−1^ at 80 °C and a peak power density of 97.8 mW cm^−2^ (2 M methanol + 6 M KOH fuel), respectively 42 and 50% higher than that of the pristine membrane. Furthermore, due to the ease of replacing quaternary ammonium ions by OH- ions via direct nucleophilic displacement and/or Hoffman elimination reactions in the presence of α, β hydrogen and α carbon atoms, a rapid decrease in conductivity was observed within the initial time range of 0 to 48 h and then gradually reached stability. The composite retains 70% ionic conductivity after immersing in 1 M KOH at room temperature for 168 h [33]. 3D hierarchical nanostructures comprised of nanoscale building blocks guaranteed a large number of active sites and surface area for conduction. Recently, Zhao et al. used 3D hierarchical flower-like layered double hydroxides (LDHs) prepared by a one-pot ethylene glycol-assisted solvothermal process, an excellent inorganic anionic conductor, to make the quaternized chitosan (QCS)/polyvinyl alcohol (PVA) blend composite with improved ionic conductivity, alkaline and mechanical stability, as well as fuel barrier properties. The main aspects for increased conductivity include interfacial contribution on the arrangement of polymer hydrophilic groups along LDH surfaces and thereby ionization and deionization of basic groups for vehicle mechanism as well as ionic hopping, transport of hydroxyl ions by basic groups via hopping, hydrogen bond formation among water molecules and hydroxyl groups on platelets, and conduction through vehicle mechanism. The power density of the fuel cell with the composite membrane exhibited a maximum power density of about 73 mW cm^−2^ which is 82.5% higher than that of the pristine chitosan membrane [8]. Layered double hydroxide coated CNT (LDH@CNTs) to quaternized chitosan/polyvinyl alcohol (QCS/PVA) blend exhibited a hydroxide ion conductivity of 47 mS cm^−2^ at 80 °C and a maximum power density of 107.2 mW cm^−2^ (2 M methanol + 5 M KOH, 80 °C) (Figure 6). It is worth mentioning that the cell voltage related to the electrode reaction and methanol permeation of the composite membrane is 0.83 V, which is greater than that of the QCS/PVA membrane (0.79 V), attributed to the fact that the hierarchical structure of LDH@CNTs can effectively hinder the methanol permeability from an anode to a cathode in the cell [6]. LDH nanosheets wrapped around silica (LDH@SiO_2_) core-shell nanocomposite modified with octadecyl dimethyl (3-trimethoxysilyl- propyl) ammonium chloride in QCS/PVA exhibited improvements in mechanical properties as well as conductivity. Direct methanol fuel cells equipped with QCS/PVA-6%QLDH@SiO_2_ produced a maximum power density of 64 mW cm^−2^ at 60 °C, and an improvement in durability owing to the excellent methanol barrier property as well as alkaline stability is also reported [34]. A bifunctional structure construction of porous substrate with a nanoarchitecture morphology encompassing a two-dimensional inorganic ion conductor, LDH attached 3D natural bacterial cellulose nanofibers (LDH@BC), served as a strong reinforcing framework as well as an ion transport medium to the quaternized chitosan (QCS) filled membrane. The composite membrane exhibited excellent alkaline stability in 2 M KOH solution at room temperature for 200 h (residual anion conductivity = 86%), owing to the steric hindrance effect of LDH@BC that can protect quaternary ammonium groups from hydroxyl ions’ nucleophilic attack to some extent, and alkaline DMFC performance of 84.2 mW cm^−2^ at 60 °C [35].

Quaternary ammonium-grafted silica coated onto carbon nanotubes (CNTs) effectively prohibits electron conduction and provides new OH^−^ conductive sites (-NR_3_^+^) to chitosan-based composites. The composite membrane with 5% functionalized CNT exhibited an ionic conductivity of 42.7 mS cm^−1^ at 80 °C and a maximum power density of 80.8 mW cm^−2^ at 60 °C in an alkaline direct methanol fuel cell. A durability test performed at a constant current of 150 mA cm^−2^ at 60 °C showed stable performance for the first 150 h, then decreased to 59 mW cm^−2^ after 240 h, followed by a continuous power loss after 300 h [3]. Quaternized chitosan-coated CNT (QCS@CNT) as filler can not only improve filler dispersion and load transfer but also endow the hydroxide ion exchange capacity of the membranes. The composite membrane of QCS@CNT in QCS prepared by Jang et al. exhibited improved mechanical properties and chemical stability. The single cell equipped with the composite membrane and operated at 60 °C outputs a maximum power density of 51.0 mW cm^−2^ and an open circuit voltage of 0.76 V, indicating good methanol barrier property owing to the distorted methanol crossover pathways of the embedded QCS@CNT [36].

AEMs with well-ordered layered structures achieve a fine balance in long-term hydroxide ion conductivity as well as chemical stability under alkaline conditions. The layered membranes prepared by Shen et al. via the combination of layer-by-layer (LBL) self-assembly membranes and a pair of electrospun poly(vinylidene fluoride-co-hexafluoropropylene) PVDF-HFP nanofiber membranes by cold pressing possessed long-term stability in hydroxide ion conductivity at different temperatures. Alternate deposition of the positively charged polyurethane (PU), quaternized chitosan (QCS), and negatively charged phosphotungstic acid (HPW) resulted in the layer-by-layer (LBL) self-assembly membranes. The external PVDF-HFP nanofiber protective layers boost the stability and the well-arranged dispersion of components through electrostatic interaction; intermolecular hydrogen bonding promotes the phase combination and diminishes the hydroxide ion conduction resistance of the AEM [37].

In Table 1, several different AEMs suitable for AEMFCs mentioned in the present review are listed based on the different functional groups, and nanomaterials present in the AEM. Despite the fact that novel chitosan based AEMs with improved conductivity as well as alkaline stability are being developed, most of them still are unable to satisfy the demand of anion exchange membrane fuel cells (AEMFCs) and have a long way to go before commercialization. Therefore, rational membrane design is still a big challenge. The preliminary results on cell performance in this review show a starting point, and optimization of the catalyst, cell temperature, and other components could give the best cell performance.

## 3. Conclusions

The durability and high production costs must be addressed for the successful commercialization of fuel cells. Nafion membranes are expensive, have low performance ratings at elevated temperatures, and contribute toward environmental pollution due to the presence of hazardous chemicals such as fluorine. Given the low cost and non-corrosive nature of biopolymers, chitosan is the best choice for replacing Nafion with an eco-friendly technology. Chitosan has amino (-NH_2_), and hydroxyl (-OH) groups that can enhance the electrochemical performance by structural modification. In this review, we have attempted to give a comprehensive overview of recent developments in modified chitosan as a potential anion exchange membrane for fuel cells. Various physical, as well as chemical, methods adopted have a significant influence on the functional characteristics of the membrane. Cross-linking improves the chemical, mechanical, as well as dimensional, stabilities of chitosan while decreasing the conductivity of chitosan membranes. Various recently published works revealed that grafting, incorporation of inorganic materials, or a combination of both before cross-linking is preferred to improve the conductivity of cross-linked chitosan-based materials. Grafting with various functional groups such as imidazolium, quaternary ammonium, benzimidazolium, as well as metal cations, etc., acts as additional ion conductive sites in chitosan membranes. One of the greatest challenges that researchers face is long-term alkaline stability and conductivity at high pH and temperatures. Various approaches, such as using modified nanoparticles, one-dimensional nanotubes, two-dimensional nanosheets with functional groups, or grafting using macromolecules, have been taken to engineer the composite membrane to overcome these challenges. Even though the advances related to the main observations above will allow chitosan membranes to be a real alternative to the presently available commercial membranes, some essential aspects such as durability and optimizing operational conditions for maximum energy efficiency remain to be addressed for large-scale application. More research is expected in the coming years to overcome these key challenges of moving from the laboratory to large-scale systems.

## Figures and Tables

**Figure 1 membranes-12-01265-f001:**
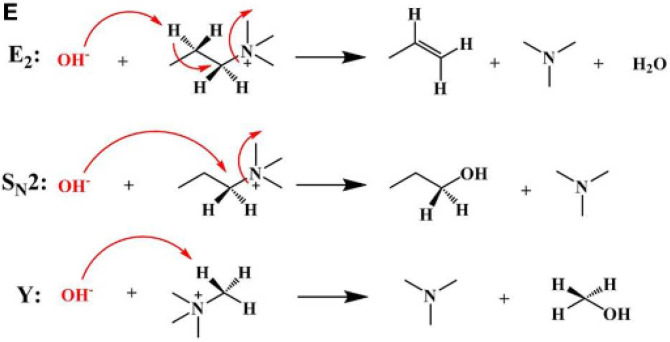
Possible degradation mechanisms of QA cations in alkaline solutions: Hofmann elimination reaction (E2), nucleophilic substitution (S_N_2), and Ylide formation (Y). Reprinted from Ref. [18].

**Figure 2 membranes-12-01265-f002:**
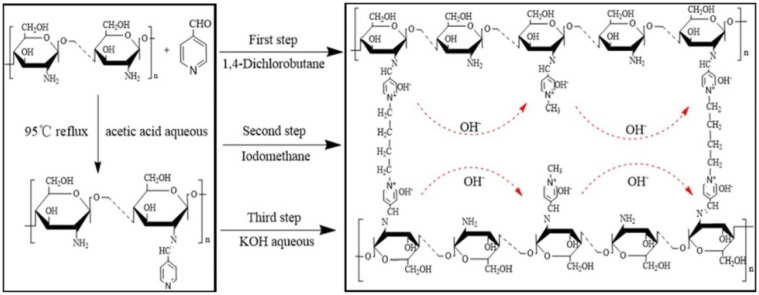
Synthesis scheme and molecular structure of CS-PY-DL_x_. Reprinted with permission from Ref. [22]. Copyright 2019 Elsevier.

**Figure 3 membranes-12-01265-f003:**
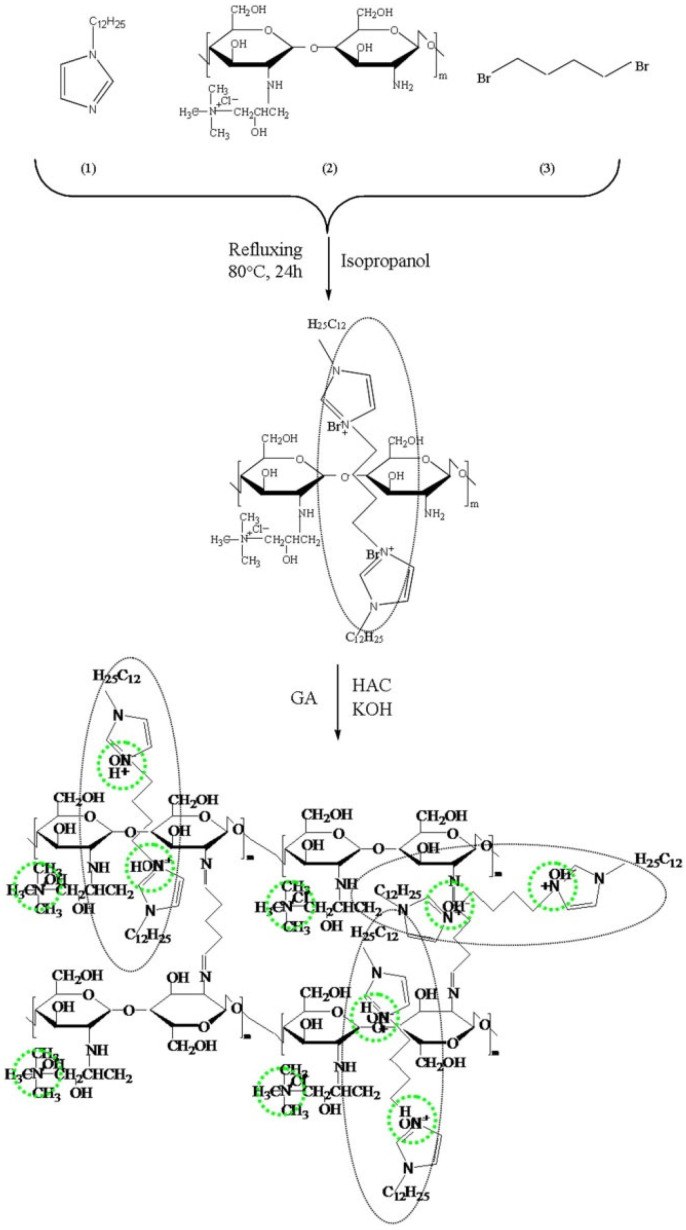
Synthetic procedure for anion exchange membranes. Reprinted with permission from Ref. [17]. Copyright 2015 Elsevier.

**Figure 4 membranes-12-01265-f004:**
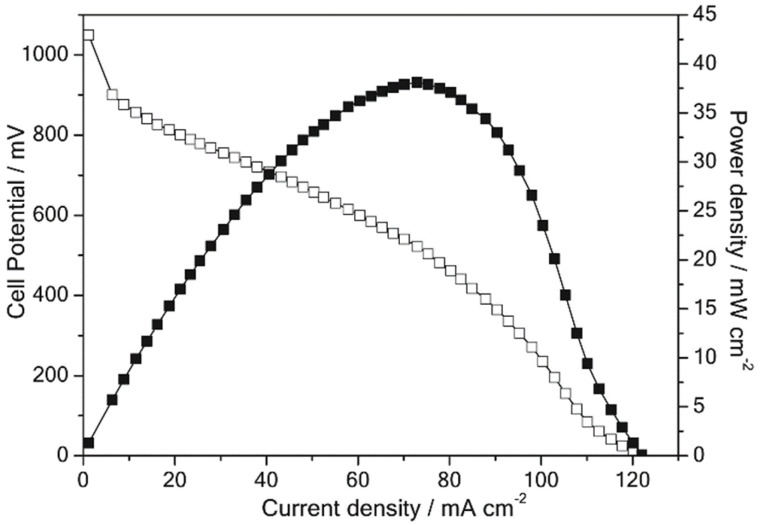
Single cell performance of CS/PAADDA/PUB (1:0.5:0.5) membrane in an H_2_/O_2_ fuel cell at 25 °C. Reprinted with permission from Ref. [25]. Copyright 2020 John Wiley and Sons.

**Figure 5 membranes-12-01265-f005:**
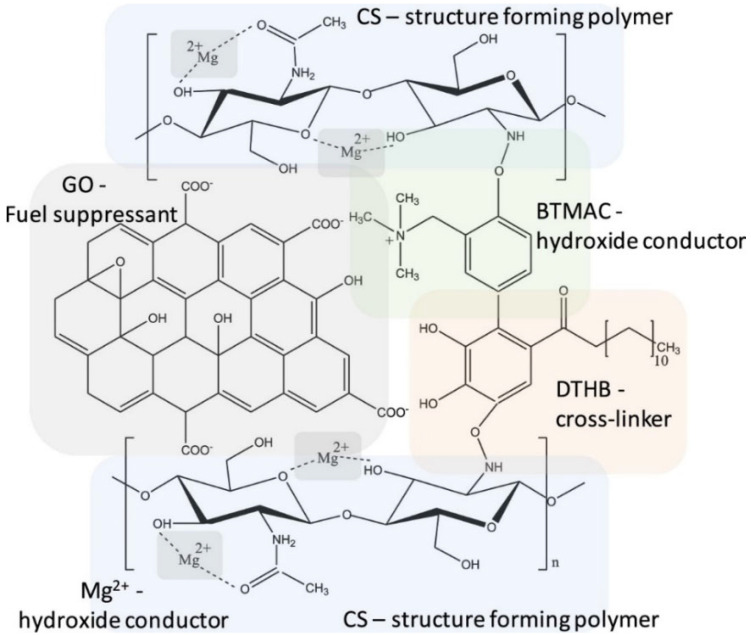
Proposed structure of the CS-Mg-GO-BTMAC-DTHB membrane and designated role of individual components. Reprinted with permission from Ref. [32]. Copyright 2019 American Chemical Society.

**Figure 6 membranes-12-01265-f006:**
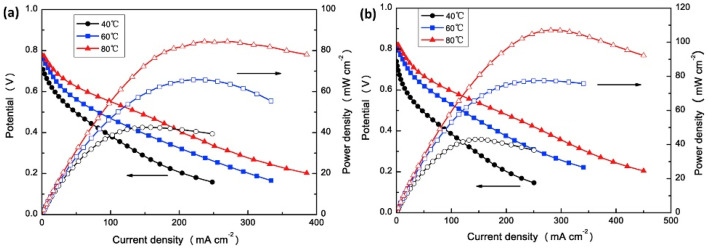
Polarization and power density curves of DMFC using (**a**) QS/PVA and (**b**) QS/PVA-1% LDH@CNTs electrolytes at different operating temperatures. Reprinted with permission from Ref. [6]. Copyright 2019 Elsevier.

**Table 1 membranes-12-01265-t001:** Properties of various AEMs suitable for AEMFCs.

Membrane Designation	Ionic Conductivity (mS cm^−1^)	Alkaline Stability (% Retention of Conductivity)	Power Density (mW cm^−2^)	Ref.
QCS/QSiO_2_@CNTs-5	42.7 (80 °C)	-	80.8 (60 °C)	[3]
QCS/PVA-1% LDH@CNTs	47.0 (80 °C)	65 % (1 M KOH, 40 °C, 192 h)	107.2 (80 °C)	[6]
D_25%_-C-K	52.18 (70 °C)	95 % (6 M KOH, 70 °C, 480 h)	-	[7]
QCS/PVA-6% LDH	38.2 (80 °C)	92 % (2 M KOH, 25 °C, 100 h)	73 (60 °C)	[8]
CS/EMImC-Co-EP	9.0 (80 °C)	100 % (8 M KOH, 85 °C, 300 h)	21.7 (RT)	[23]
QCS/ QSiO_2_@PVDF	41.0 (80 °C)	-	98.7 (80 °C)	[27]
CS/Mg(OH)_2_/GO/BTMAC	142.5 (40 °C)	-	68.6 (80 °C)	[32]
QCS/PVA-5%-B-LDH	35.7 (80 °C)	70 % (1 M KOH, RT, 168 h)	97.8 (60 °C)	[33]
QCS/0.15-LDH@BC	42.5 (80 °C)	86 % (2 M KOH, RT, 200 h)	84.2 (60 °C)	[35]

## Data Availability

Not applicable.

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
