# Peer review of "A Review of Recent Chitosan Anion Exchange Membranes for Polymer Electrolyte Membrane Fuel Cells"

_membranes, 2022, doi:10.3390/membranes12121265_

Round 1

Reviewer 1 Report

 After reading it carefully, in my opinion, it was recommended for the acceptance in the journal Membranes. My suggestion was that some latest papers (doi: 10.1016/j.eurpolymj.2020.110160; 10.1016/j.ijhydene.2020.03.175) about the application of chitosan in proton exchange membrane should be mentioned in the section of introduction.

Author Response

The authors thank the reviewer for the critical review. Both references have been cited in the introduction (4,5) (highlighted in blue colour).

Reviewer 2 Report

The current mini-review explained the chitosan-based anion exchange membrane for the PEMFC applications. The effect grafting of various counterions for the improvement in ionic conductivity, and chemical and mechanical stability of the AEMs. The review is very concise and limited to the synthesis of membranes. Major important factors for AEMs need to be taken care of in the review, and a few of them are listed below,

1) Degradation mechanism of AEM is still a big question. The author must include the explanation regarding the same with proper/probable pathways involved in it.

2) The mechanism of ionic conduction in the explained composites needs to be discussed explicitly.

3) The explanation related to the actual fuel cell performance characteristic of the chitosan-based AEM must be provided.

4) The Pros and Cons of the explained synthesis procedure must be discussed.

5) Provide explicit information on the future prospects of the Chitosan-based AEMs

Author Response

1. Degradation mechanism of AEM is still a big question. The author must include the explanation regarding the same with proper/probable pathways involved in it.

Ans: Authors thank the reviewer for the critical review. The response to comments is highlighted in blue colour.

The main degradation mechanisms including direct nucleophilic substitution (SN2) or Hofmann elimination reaction (E2) or Ylide formation (Y), etc, are incorporated in the revised manuscript.

2. The mechanism of ionic conduction in the explained composites needs to be discussed explicitly.

Ans: The possible ion conduction mechanism (diffusion, hoping, etc.) in various composites are incorporated in the revised manuscript.

3. The explanation related to the actual fuel cell performance characteristic of the chitosan based AEM must be provided.

Ans: Fuel cell performance of the various membranes available are discussed in detail in the revised manuscript.

4. The Pros and Cons of the explained synthesis procedure must be discussed.

Ans: Pros and Cons of the various synthesis procedures are mentioned in the revised manuscript.

5. Provide explicit information on the future prospects of the Chitosan-based AEMs

Ans: Prospects of Chitosan-based AEMs are listed in the concluding section of the revised manuscript.

Reviewer 3 Report

Dear authors, 

This manuscript made a review of the performance of chitosan-derived membranes for AEMFCs in the 2015-2022. Various strategies such as the preparation of structurally modified hydrophilic/hydrophobic chitosan or creating membranes based on organic/inorganic hybrids are included.

This is a well written and organized review, and can provide a good reference for those who would like to do some research in chitosan anion exchange membranes for polymer electrolyte membrane fuel cells. I suggest it to be published without revision.

Author Response

The authors thank the reviewer for the critical review and acceptance of our manuscript.

Round 2

Reviewer 2 Report

The revision work is satisfactory and can be considered.